# A Framework for Cloud-Based Spatially-Explicit Uncertainty and Sensitivity Analysis in Spatial Multi-Criteria Models

Christoph Erlacher [1,2,*], Karl-Heinrich Anders [2], Piotr Jankowski [3,4], Gernot Paulus [2] and Thomas Blaschke [1]

1    Department of Geoinformatics, University of Salzburg, 5020 Salzburg, Austria; thomas.blaschke@sbg.ac.at
2    Department of Engineering & IT, Spatial Information Management, Carinthia University of Applied Sciences, 9524 Villach, Austria; K.Anders@fh-kaernten.at (K.-H.A.); G.Paulus@fh-kaernten.at (G.P.)
3    Department of Geography, San Diego State University, San Diego, CA 92182-4493, USA; pjankows@sdsu.edu
4    Institute of Geoecology and Geoinformation, Adam Mickiewicz University, 61-680 Poznań, Poland
*    Correspondence: christoph.erlacher@stud.sbg.ac.at or C.Erlacher@fh-kaernten.at; Tel.: +43-(0)5-90500-2245

**Abstract:** Global sensitivity analysis, like variance-based methods for massive raster datasets, is especially computationally costly and memory-intensive, limiting its applicability for commodity cluster computing. The computational effort depends mainly on the number of model runs, the spatial, spectral, and temporal resolutions, the number of criterion maps, and the model complexity. The current Spatially-Explicit Uncertainty and Sensitivity Analysis (SEUSA) approach employs a cluster-based parallel and distributed Python–Dask solution for large-scale spatial problems, which validates and quantifies the robustness of spatial model solutions. This paper presents the design of a framework to perform SEUSA as a Service in a cloud-based environment scalable to very large raster datasets and applicable to various domains, such as landscape assessment, site selection, risk assessment, and land-use management. It incorporates an automated Kubernetes service for container virtualization, comprising a set of microservices to perform SEUSA as a Service. Implementing the proposed framework will contribute to a more robust assessment of spatial multi-criteria decision-making applications, facilitating a broader access to SEUSA by the research community and, consequently, leading to higher quality decision analysis.

**Keywords:** Spatially-Explicit Uncertainty and Sensitivity Analysis; parallel and distributed computing; SEUSA as a Service; spatial cloud computing; microservices; Spatial Multi-Criteria Decision Analysis; Python–Dask; gRPC; RasDaMan; Kubernetes

## 1. Introduction

Spatial uncertainty and sensitivity analysis is a crucial step in Spatial Multi-Criteria Decision Analysis (S-MCDA) to verify mathematical models' robustness and stability, incorporating the existing uncertainties. S-MCDA applications for natural hazard risk assessments, landscape assessment, identification of land-use strategies for sustainable regional development, or habitat suitability in the context of environmental protection very often do not provide detailed information about the robustness and uncertainty of the results. These applications currently lack not only quantifiable measures of solution robustness, but also the estimates of spatial distribution of uncertainty at any given location that affects the model outcomes. Incorporating uncertainty and sensitivity analysis in a modeling procedure leads to a significant increase in the quality of the analysis and, consequently, better decisions. However, running this type of analysis is a complicated and highly time-consuming computational process, especially for huge raster datasets. Therefore, this article focuses on the design and development of a framework for a scalable and adaptable Cloud-based Spatially-Explicit Uncertainty and Sensitivity Analysis (C-SEUSA).

The combination of Decision Support Systems with Geographic Information Systems (GIS) is known as Spatial Decision Support Systems (SDSS). It can provide powerful tools to

support various application domain experts during the decision-making process. SDSS is a computer-based system that incorporates non-spatial and spatial data, GIS-based analysis and visualization functions, and decision models regarding specific domains, to "facilitate the evaluation of solution alternatives and the assessments of their trade-offs" [1] (p. 64). Vector data (e.g., points, polylines, or polygons) or raster data can represent location alternatives in spatial decision-making and can be described by spatial and non-spatial characteristics. Spatial characteristics of alternatives relate to geographical location and spatial relations (e.g., proximity, adjacency, and contiguity). The three-phase model rational decision-making process suggested by Simon [2] consists of the intelligence phase (problem definition, including evaluation criteria and constraints), design phase (data collection and model construction), and choice phase (where location-based choices are made). Both Multi-Criteria Analysis (MCA) for non-spatial applications and S-MCDA pass through these phases. S-MCDA has been prominently represented in GIScience [3–5]. According to Malczewski and Jankowski [3], the number of peer-reviewed manuscript publications in scientific journals regarding S-MCDA has been increasing exponentially, owing to the ongoing progress in GI-technologies, the diversity of GIS–MCA applications, and the availability of spatial data. One important and critical part of the choice phase in SMCA is the Sensitivity Analysis (SA). SA verifies the robustness and stability of the model results with respect to existing uncertainties and indicates a key ingredient concerning the quality of a model-based study [6]. Uncertainties can refer to the problem structure, combination rules, criterion values, and criterion weights caused by inaccuracy, differences in human judgments, or measurements errors [3,7–9]. Saltelli and Annoni [6] and Ferretti et al. [10] observed that the majority of scientific contributions regarding SA have involved either local or one factor-at-a-time methods with more robust global approaches to sensitivity analyses are slowly gaining recognition. Global SA (GSA) approaches, such as variance-based SA, represent an attractive option for spatial models, because of the multi-parametric and non-linear nature of many spatial decision problems [11–14]. As stated by Lilburne and Tarantola [12], intricate non-linear models require a high number of simulation runs in order to obtain more precise sensitivity estimates. This, in turn, has a negative effect on the analysis runtime. In particular, sequential solutions performing a variance-based Spatially-Explicit Uncertainty and Sensitivity Analysis (SEUSA) are often limited by a compromise between the number of simulation runs and the quality of the model sensitivity measures [15,16]. Erlacher et al. [17] and Erlacher et al. [18] presented GPU-based parallelization approaches to accelerate the generation of the suitability surfaces, which constitute an input for the uncertainty and sensitivity analysis for each pixel location. Erlacher et al. [19] introduced a cluster-based parallel and distributed solution for large raster datasets that focuses on a variance-based Spatially-Explicit Uncertainty and Sensitivity Analysis for a land-prioritization model. Although the achieved solutions showed a considerable acceleration, even for large raster datasets over the sequential processing approach, it also revealed the computing and storage requirements that could easily exceed local clusters' capacity limits.

The computation effort in SEUSA depends on (1) the complexity of the model; (2) the number of evaluation criteria; (3) the number of decision alternatives (4); the number of simulation runs to obtain reliable sensitivity indices; (5) various spatial, spectral, and temporal resolutions; and (6) spatial decision support applications. For example, earth observation satellites, unmanned aerial vehicles, airborne laser scanners, and terrestrial mobile mapping systems generate terabytes of raster datasets [20–23]. Consequently, the primary objective of this research is to design and develop a scalable and adaptable SEUSA framework applicable to a wide range of S-MCDA models.

In the following section, we introduce the extended parallel and distributed SEUSA approach based on Python–Dask. Furthermore, we present a prototype implementation allowing various client applications to communicate with the parallel SEUSA approach, thus enhancing its accessibility and usability. In Section 3, we discuss a migration from a parallel to distributed approach in order to make SEUSA applicable for complex S-

MCDA use cases, incorporating massive raster datasets. The proposed solution involves cloud-based architectures and applications as well as requirements such as geoprocessing and tiling services, cloud storage for geospatial information, and GIS as a service to provide data preparation, analysis, and visualization functionalities. The paper concludes with a reflection of the proposed framework and provides prospects concerning further development of the cloud-based SEUSA approach.

## 2. Methodology—Parallel and Distributed SEUSA Approach

### 2.1. SEUSA: Python–Dask

Erlacher, Desch, Anders, Jankowski, and Paulus [19] presented a parallel and distributed approach based on Python–Dask to shorten the processing time generating the suitability surfaces stack. As stated by Hadjidoukas, et al. [24] (p. 3), the Dask framework is message-queue-based and follows a client–scheduler–worker approach that mainly targets cloud computing environments. Dask supports libraries such as NumPy, Pandas, and Scikit-Learn, and provides a responsive real-time dashboard, a fault-tolerant behavior, and different types of schedulers [25,26]. This stack incorporates several millions of pixel locations and a hundred thousand simulation realizations. Each simulation result represents the performance of the location alternatives (pixel locations) for a sample of the criterion weight values. The creation of the weight samples (SAM Files) is implemented by applying Sobol's quasi-random experimental design [27] to compute the radial weight matrix. The Python–Dask-based solution is applicable for local clusters that consist of a set of workstations. The master scheduler is responsible for scaling and for distributing the workload among the nodes. Meta-information regarding the capacities (e.g., number of threads per core, available memory) of the worker nodes, and the precalculated workload for all processing operations, are relevant to guarantee a balanced distribution of the datasets. All nodes of the local cluster have direct access to the raster input datasets over the Network File System (NFS). The number of simulations affects the Dask array's chunk sizes, where the first index identifies a criterion map and the remaining two indices refer to the row and column of a specific criterion map. A *dask.array* consists of smaller NumPy arrays (blocks), which allows performing the processes on arrays larger than the memory available for worker nodes. All functions, such as the in-memory generation of suitability surfaces and the average and standard deviation map creation, are mapped to the blocks. This approach only incorporates the average and standard deviation map computation in parallel. Additional functions, which are necessary to generate the first-order sensitivity measures (Si indices) and total-order sensitivity measures (STi indices) for each criterion and location in the raster datasets, are migrated and mapped to each block of the *dask.array*. The mapped functions constitute future objects and contain pointers to respective arrays. This enhancement represents a more effective integration of the sensitivity surface generation in parallel and results in a computational performance increase compared with the sequential solution. All computations of the SEUSA approach for each block are executed by calling the NumPy asarray() function. The intermediate stack of suitability surfaces incorporates a subarea (first two dimensions) of the study site, where the third dimension indicates the suitability values for each simulation run. The number of suitability surfaces (Equation (1)) depends on the chosen weight sample size (N) and the number of criteria (i):

$$R = (i + 2) * N \tag{1}$$

Figure 1 compares the needed working memory to generate the suitability surfaces depending on the number model runs. This example incorporates five criteria and approximately 13 million pixel locations. The total amount of working memory for the suitability surface of data type Float32 (single precision) and Float64 (double precision) is given for each number of simulation runs: 9856, 19,712, 39,424, 78,846, and 157,696. The higher the number of model runs, the smaller the block size. As a result, with 157,696 simulations, the calculation time increases drastically, and local clusters reach their limit [19]. This limit

is reached earlier, especially with high-resolution raster data and multispectral imagery covering a large area.

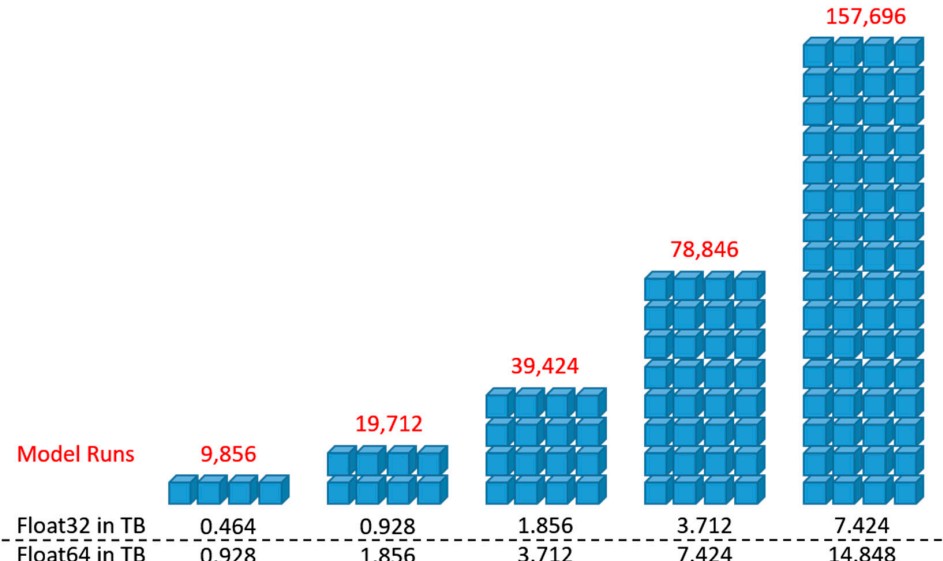

**Figure 1.** Spatially-Explicit Uncertainty and Sensitivity Analysis (SEUSA) memory consumption. Comparison of the working memory consumption to generate the suitability surfaces, depending on the number of model runs.

The intermediate suitability surfaces represent the input of the Uncertainty and Sensitivity Analysis (USA). For each block, the uncertainty surfaces, average map, standard deviation map, and the sensitivity surfaces, including the first-order and total-order maps, are stored into arrays that cover a specific area of the whole study site. Afterward, all surfaces in the working memory are deleted to avoid a memory overload for subsequent workloads. As long as not all blocks are processed, the scheduler distributes the workload among the local cluster nodes. Figure 2 illustrates the simplified representation of the parallel and distributed SEUSA approach's simplified representation based on Python–Dask. For the calculation of the optimal block sizes, further tests are necessary concerning additional application scenarios. The number of criteria, the decision rule used, and the number of pixels within the project area are the essential factors that are decisive for the optimal block size.

*2.2. SEUSA: Middleware*

The presented parallel and distributed SEUSA approach based on Python–DASK is not directly accessible for client applications. Therefore, another essential aspect of the approach needs to be addressed in order to make SEUSA more accessible to a broader user community. The integration of a middleware represents an opportunity to connect various GIS-based client applications with multi-core and cluster-based SEUSA methods. Middleware is an umbrella term for reusable software, including patterns and frameworks to facilitate applications' functional requirements and the incorporated operating systems, network protocol stacks, and databases [28,29]. Simply put, the middleware acts as a software layer between the clients and the application. The term middleware does not pertain to a specific type of software. It depends on the kind of application, such as Message Oriented Middleware (MOM), Remote Procedure Calls (RPC), Database Middleware, Application Programming Interface (API), Object Middleware, Transaction Processing Middleware (TP), or Device Middleware. Various aspects of middleware design and implementation pertain to the performance of sending information, portability between different platforms, interoperability, security issues, and effortless implementation of additional functionalities. This research refers to RPC based on gRPC to connect GIS-based client applications with the parallel and distributed SEUSA approach. The gRPC is a

high-performance, open-source universal RPC framework developed by Google that can operate in any environment and supports languages such as C++, Java, Python, Go, Ruby, Dart, and C#. This framework uses a protocol buffer as the Interface Definition Language (IDL) and its underlying message interchange format. Protocol buffers are language neutral and platform neutral, representing an extensible way of serializing structured data, and relate to a binary message format that provides faster parsing and is less storage-intensive. Due to the mentioned advantages, gRPC is widely used in different areas [30–32]. The client stub acts as a local representative of the server on the client-side and vice versa. The calling process (client) and the procedure body (server) refer to the same interface (proto files). Both stubs rely on a communication subsystem to exchange messages. The client and server stubs use a naming service, which supports the client to locate the server. The calling process or thread (client) must be blocked while waiting for the procedure to return. A daemon thread is waiting for incoming messages on a predefined port (direct execution).

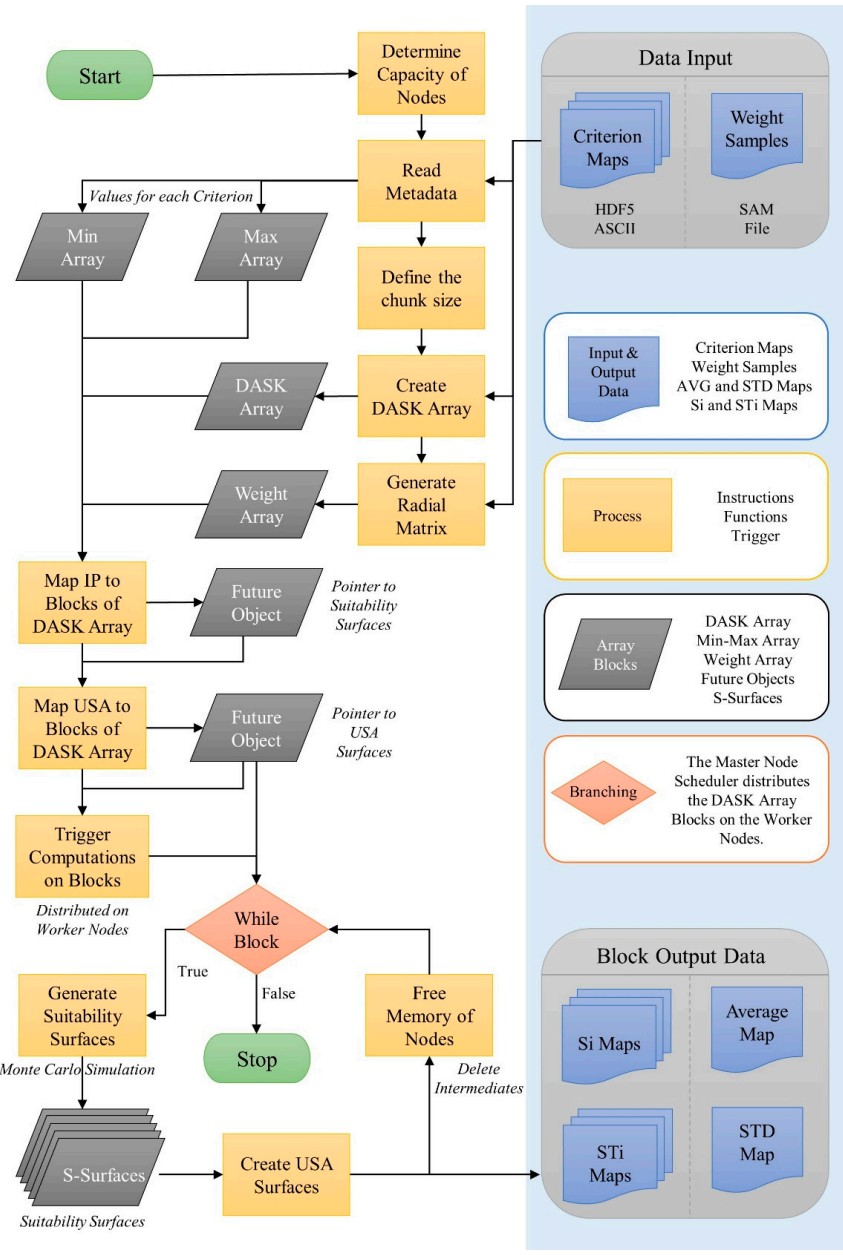

**Figure 2.** SEUSA–DASK workflow. Parallel and distributed SEUSA approach based on Python–DASK.

Figure 3 illustrates the SEUSA *.proto* file that comprises the service definition *MyServiceNdArray* incorporating the procedure *GetSEUSASurface* that takes the Request*NDArray* parameter from the client and returns the *ReturnNDAarray* message from the server. The NumPy arrays for input and output are of data type bytes, and the supported decision rules are of type enum. The generated Python code file for the service definition include the following:

(1) the Stub, which can be used by the client application to invoke remote procedure call;
(2) the Servicer, which defines the interface for the implemented services; and
(3) the Servicer _to_server function, which adds the Servicer to the grpc.Server.

```
// The SEUSA Service Definition
service MyServiceNdArray{
  // The SEUSA Remote Procedure Call
  rpc GetSEUSASurface(RequestNDArray) returns (ReturnNDArray){}
}

// input gRPC Message
message RequestNDArray {
  bytes nd_input_array = 1;      // NumPy Array [criterion map, row, column]
  bytes nd_weight_array = 2;     // NumPy Array [samples, weight values]
  enum DecisionRule {
    WLC = 0;
    IP = 1;
  }
  DecisionRule ds = 3;           // Type of Decision Rule
}

// output gRPC Message
message ReturnNDArray {
  bytes nd_seusa_array = 1;      // NumPy Array [surface map, row, column]
}
```

**Figure 3.** The proto file. A simple SEUSA proto example.

A second Python code file incorporates special descriptors for the proto file and all messages, enumerations, and fields.

The service interface incorporates the function, which invokes the Dask implementation and performs the service's actual work. The running gRPC server waits for requests from clients and transmits responses. Our prototype has been implemented for a multi-core workstation with the open-source GI-System QGIS. The QGIS client gRPC runs in the background to keep all other functions in QGIS available for users. The gRPC channel (secure or insecure) connects the gRPC server to a specified host and port. According to the interface definition, the client stub uses the channel to submit and receive the serialized messages (input and output). Finally, the converted raster datasets comprise the SEUSA surfaces and are directly available for the users in QGIS. Figure 4 provides a simplified schematic representation of the gRPC and parallel SEUSA approach workflow.

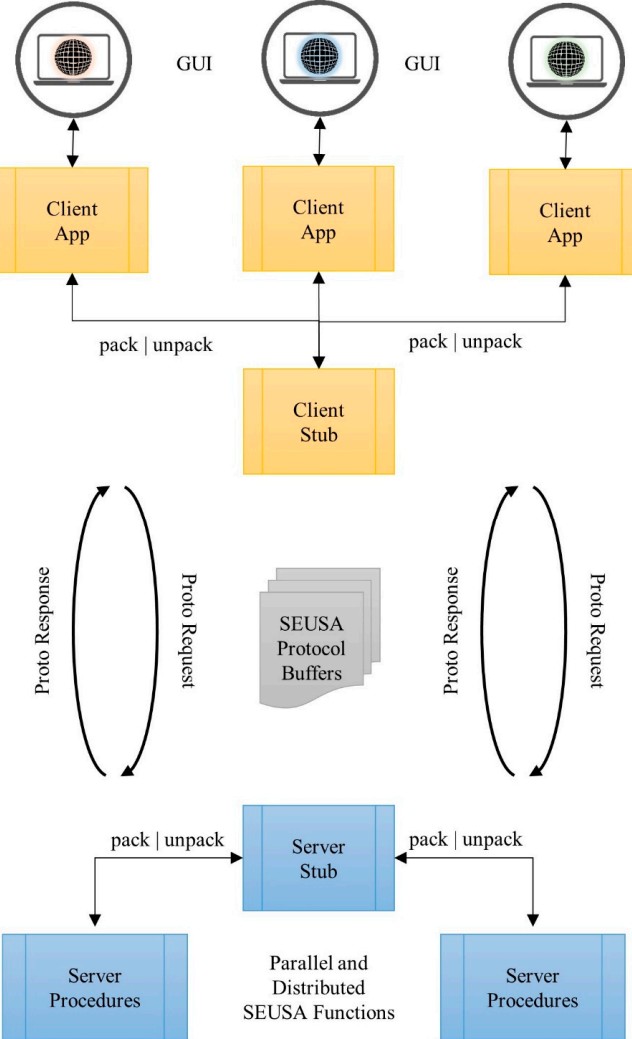

**Figure 4.** gRPC–Dask workflow. Connection of GIS-based client applications with the parallel and distributed SEUSA approach by using gRPC.

## 3. SEUSA to Cloud—Design of the Framework

This section provides an overview of the theoretical background, architectures, and spatial case studies relevant to designing the cloud-based framework for SEUSA. The requirements represent a crucial step for designing and developing the cloud-based SEUSA approach and are illuminated and described in this section. They refer to the following aspects:

1. General needs for the cloud migration;
2. SEUSA as a Service to facilitate access for user communities;
3. parallel and distributed computing issues;
4. tiling services relevant to perform computations and facilitate map representation; and
5. requirements concerning cloud storage to provide high data availability and reliability for exchanging information between applications.

The design of the cloud-based framework of the SEUSA approach highlights the architectural design that represents the basis for the implementation of the application.

### 3.1. Theoretical Background

Cloud-based applications and services are used in everyday life by many people, such as film streaming, online banking, and social networks. The term cloud computing was coined in late 2007 as a new computing paradigm that provides reliable, customized, and Quality of Service (QoS) guaranteed dynamic computing environments for end-users [33].

Wang, von Laszewski, Younge, He, Kunze, Tao, and Fu [33] (p. 139) provide the following definition for cloud computing: "A computing Cloud is a set of network-enabled services, providing scalable, QoS guaranteed, normally personalized, inexpensive computing infrastructures on demand, which could be accessed in a simple and pervasive way." According to the National Institute of Standards and Technology, U.S. Department of Commerce, cloud computing represents a model that facilitates ubiquitous, convenient, on-demand network access to a shared pool of configurable computing resources (e.g., networks, servers, storage, applications, and services), which can be rapidly provisioned and released with minimal management effort or service provider interaction [34]. A collection of more than 20 early cloud definitions are provided by Vaquero et al. [35].

The three main service model types are Infrastructure-as-a-Service (IaaS), Platform-as-a-Service (PaaS), and Software-as-a-Service (SaaS) [34–37]. (1) IaaS represents the most flexible cloud model, where the cloud providers host the infrastructure resources, such as servers, storage, computing resources, and network hardware for the end-users. Users can deploy and run any desired software, including operating systems and their applications, without managing the underlying cloud infrastructure. The infrastructure providers can split, assign, and dynamically resize these resources according to the end-users' needs [35] (p. 52). Examples of this service cloud model are Amazon Elastic Compute Cloud (EC2), Microsoft Azure IaaS, and Google Compute Engine. (2) PaaS represents a higher level of service than IaaS and offers platform services for the software developers for implementing applications [38]. Providers are responsible for controlling and maintaining the cloud infrastructure, such as the operating system, networks, servers, and storage. Additionally, they offer Application Programming Interfaces (API), integrated development environments that support various programming languages, and dashboards to monitor the applications. Examples of PaaS are Google App Engine, Microsoft Azure Cloud Service, and AWS Elastic Beanstalk. (3) SaaS provides end-users the opportunity to access specific web-applications over the internet but allows the least flexibility regarding the environment and hardware over which the services are running [39]. The service hosts the application and manages security issues, updates of the used software components, and the software's availability and performance utilization. Examples of SaaS are ArcGIS Online [38,40] and Google Apps. Agrawal and Gupta [41] provide a compact comparison between cloud computing delivery models, including the characteristics and examples for each cloud model. In addition to the first three service model types (IaaS, PaaS, and SaaS), Data as a Service (DaaS) are essential to geospatial sciences [38] (p. 310). DaaS represents a delivery mechanism for geospatial data repositories allowing users to discover, retrieve, and upload data from remote machines over the network.

According to Mell and Grance [34], the three cloud service models can be deployed either as a private, public, community, or hybrid cloud. (1) Private clouds only provide access for a specific organization (e.g., institution or company) and do not share their service outside their organization. (2) In a public cloud, multiple organizations can access the same infrastructure offered by a service provider. The cloud service provider manages and maintains the security in the cloud service to prevent unauthorized access. The payment method usually refers to the pay-as-you-go model [39]. (3) In community clouds, the infrastructure is provided only to a particular group of users from organizations that share special requirements and common concerns (e.g., government organizations, pharmaceutical companies, insurances, or financial institutions). The advantage of community clouds over private clouds is that the former reduces capacity needs by sharing resources, reducing overhead, and reducing costs. (4) Hybrid clouds represent a combination of various deploying models (e.g., private, public, and community clouds), which allows communication between different services through a proprietary software. Such cloud models are beneficial to organizations for two reasons. They are cost-efficient because of shared resources. Moreover, they promote better control and higher security level of private and sensitive data. Cloud computing incorporates five essential characteristics,

including on-demand self-service, broad network access, resource pooling, rapid elasticity, and measured service.

According to Yang, Goodchild, Huang, Nebert, Raskin, Xu, Bambacus, and Fay [38], challenges in geospatial sciences regarding information technology in the twenty-first century refer to data intensity, computing intensity, concurrent access intensity, and spatiotemporal intensity. Satellites [42], unmanned aerial vehicles [43], terrestrial mobile mapping systems [44], autonomous cars [45], mobile phones [46], and location and human sensors [47] collect, use, and share an immense amount of geospatial data on a daily basis, at multiple locations, at different scales, and in various data formats. This results in significant challenges in terms of the organization and management of data content, data format and data services, data structure and algorithms, data processing, data distribution and identification, as well as data access and utilization [38,48]. On the one hand, cloud storage services like Amazon Simple Storage Service (Amazon S3) or Amazon Elastic Block Store (EBS) provide scalable and foul tolerant object storages services and block storage services to address challenges concerning the storage and access of huge geospatial datasets. On the other hand, transmitting and hosting large amounts of spatial data on the cloud is expensive [23,49]. Additionally, transferring a huge amount of data has a negative impact on performance, and therefore data compression algorithms are necessary to reduce the data size beforehand [23,50]. Li et al. [51], for example, presented an efficient network transmission model that supports multiple data encoding methods such as Geography Markup Language and GeoJSON, and compression techniques like LZMA and DEFLATE. Moving functions (source code) to data represents a promising solution, but needs further research concerning code migration for distributed heterogeneous systems [52] (p. 219).

As stated by Li, Gui, Hofer, Li, Scheider, and Shekhar [52] (pp. 208-209), for a web-based distributed geospatial information processing ecosystem, everything is encapsulated as a service (XaaS). For example, such services might refer to data processing, visualization, knowledge management, model chaining, data mining, sensor web, collaboration, or agent-based services [53–58]. In Section 3.3, further relevant aspects for C-SEUSA concerning cloud-based architecture, such as container virtualization and serverless computing, tiling services, and cloud storage, are illuminated for the framework's design.

### 3.2. Requirements

The requirements (Table 1) for developing the cloud-based SEUSA approach are classified according to priority into high, moderate, and low. The requirements referring to a high priority focus on the leading essential aspects concerning the migration of the parallel and distributed SEUSA solution based on Python–Dask. The communication between microservices for exchanging information and spatial raster datasets, cloud storage supporting spatial data, and services providing aligned and arbitrary tiling also represents high priorities. A web-based portal to store, query, analyze, and visualize all datasets relevant to spatial decision support indicates a high priority requirement. Requirements relating to moderate priority include providing additional functionalities for preparing criterion maps and further implementing decision rules. The development of a decision wizard for the SEUSA approach and the implementation of different sampling strategies represent low priority requirements.

**Table 1.** SEUSA to cloud requirements. Requirements for the cloud-based development of the parallel and distributed SEUSA approach are divided into general requirements, SEUSA as a Service, parallel and distributed computing, tiling services, and cloud storage.

| Priority | SEUSA to Cloud Requirements |
| --- | --- |
| **(1) General Aspects** | |
| High | All provided services should automatically scale up or scale down according to the given workloads. |
| High | Interfaces between the presentation, application, and data layer for<br>• exchanging information and data,<br>• triggering events and methods, and<br>• facilitating communication between microservices<br>have to be considered. |

**Table 1.** *Cont.*

| Priority | SEUSA to Cloud Requirements |
|---|---|
| Moderate | Different aspects concerning the deployment of the cloud service models, such as private, public community, or hybrid cloud, should be incorporated to cover various user communities' needs. |
| Low | A virtual cloud network that provides a secure managed network for cloud services, where managed firewalls are deployed, and security assessment is conducted in advance, should be integrated. |
| **(2) SEUSA as a Service** | |
| High | The parallel and distributed SEUSA methods should be accessible utilizing a Web-GIS application to allow the user communities to store, query, analyze, and visualize the spatial datasets. |
| High | User communities of the SEUSA framework should have the opportunity to upload their use cases that incorporate spatial- and non-spatial datasets (raster or vector data, weight samples, the type of the decision rule) and retrieve results of SEUSA computations. |
| Low | The development of a decision wizard for the SEUSA framework should be designed as Workflow as a Service that facilitates the application's usability. |
| **(3) Parallel and Distributed Computing** | |
| High | The parallel version of the SEUSA approach represents the most time- and memory-intensive part of the proposed implementation. This approach is based on Python–Dask, and therefore suitable cloud architectures have to be identified for the development. |
| Moderate | The integration of different standardization methods, such as S- and J-shaped functions, should be implemented for preparing the criterion maps, which allows for covering a large number of use cases and offers flexible expert specifications. |
| Moderate | Additional decision rules like Ordered Weighted Averaging (OWA) and Analytical Hierarchy Process (AHP) that can generate a suitability surface for each model run, extend the applicability for various application domains. |
| Low | The SAM files are currently used to create weight samples. Hence, the creation of the weight samples should be integrated directly into the application. Furthermore, the integration of additional weighting and sampling methods should be considered. |
| **(4) Tiling Service** | |
| High | Map caching for presenting the spatial information and tiling services for the parallel and distributed computing of SEUSA has to be investigated to increase the Web-GIS applications' speed. |
| High | For the raster datasets, aligned tiling (chunking), where all chunks have the same size, and arbitrary tiling, or where tiles consist of sub-areas of different sizes, have to be supported. Significantly, local multi-criteria evaluation approaches require arbitrary tiling services to calculate criterion weights for local neighborhoods [11,15,59]. |
| **(5) Cloud Storage** | |
| High | Geospatial information storage requirements can range from a few gigabytes of data up to terabytes or petabytes of data, particularly for high-resolution multispectral or hyperspectral images. Therefore, scalable cloud-based storage services to host and share a large volume of spatial data have to be considered. |
| Moderate | Cloud archive storage should be incorporated for data that is not frequently accessed and can be used for data recovery. |

### 3.3. Architectural Design

This section focuses on developing the SEUSA approach based on Python–Dask and the geoprocessing operations for preparing the criterion maps in a cloud environment. As stated by Hollaway et al. [60] (p. 8), a larger volume of data, complex models, or data at a finer spatial scale requires the cloud's computational elasticity, which can be accomplished by distributed computing resources like Dask and Spark clusters. There are various ways to use DASK for cloud-based applications, such as a managed Kubernetes service and Helm, vendor-specific services such as Amazon ECS and Dask Cloud Provider, or a managed Yarn service (e.g., Amazon Elastic MapReduce, Google Cloud DataProc, and Dask-Yarn). The Kubernetes, initially developed by Google, is an open-source system for automating deployment, scaling, and the management of containerized applications. Helm is the package manager for Kubernetes, where Helm Charts helps define, install, share, and upgrade the majority of complex applications based on Kubernetes. The Helm Charts requires instructions stored in YAML files that incorporate dependencies and resources stemming from how Kubernetes applications are built. YAML represents a data-serialization language readable for humans. Dask-Yarn applications deploy on Apache Hadoop YARN clusters and provide a simple interface that rapidly starts, scales, and stops Python–Dask containerized applications within a cluster. Additionally, DASK-Yarn utilizes the Pythonic library Skein that facilitates the deployment of applications on Apache Hadoop YARN.

Container as a Service (CaaS) represents a cloud-based service architecture, which provides the opportunity to upload, organize, run, scale, and manage containers utilizing container-based virtualization. The containers virtualize an Operating System (OS) so that various applications are executed on just one instance of an OS, whereas virtual-machine deployments virtualize a machine. Therefore, application containers are often designated as lightweight virtual machines Huang et al. [61]. Containers incorporate their own file system, CPU, memory, process space, decoupled from the underlying infrastructure, and therefore are easier to transfer across clouds and OS distributions than virtual machines. Iosifescu-Enescu et al. [62] proposed a cloud-based architecture for geoportals incorporating microservices to connect applications for geoprocessing operations that follows a serverless computing model. The serverless model represents a cloud computing model that offers to manage the allocation and provisioning of servers and automatically scale up and down to the given workload. Containers supporting a serverless computing model are stateless and do not incorporate any persistent storage (e.g., user settings and preferences, temporary storage, environment variables, files, and databases). Kubernetes supports both stateless and stateful cloud-based applications and is platform-agnostic. Therefore, developers can deploy their Kubernetes-based applications on various cloud platforms such as AWS, OpenStack, GCP, OpenShift, Microsoft Azure, or private data and computing centers. Stateless cloud applications require serverless storage such as object storage (e.g., Azure Blob, and Amazon S3, Swift, and Google Cloud Storage) and application memory cache. As recommended by Iosifescu-Enescu, Matthys, Gkonos, Iosifescu-Enescu, and Hurni [62] (p. 10), all operationally-critical resources should be provided by a high-speed data storage like serverless caching services (e.g., Azure Redis Cache, Redis Cloud, Amazon ElastiCache, and Nutanix Acropolis) to increase the performance. In combination with the Kubernetes, Dask provides adaptive deployment methods, which are particularly helpful for interactive workloads. The Dask scheduler uses the external resource scheduler from Kubernetes, incorporating generated auto-scaling rules for dynamically launching and closing workers depending on the workload and the compute utilization.

As defined in Section 3.2, aligned and arbitrary tiling of raster datasets are an essential requirement, especially for local multi-criteria-evaluation approaches. An Array Database Management System (DBMS), such as RasDaMan, SciDB, and TileDB [63], allows retrieving tiles from multi-dimensional arrays using an Array Query Language (AQL) [64]. RasDaMan represents an OGC reference implementation and provides functionalities for storing, processing, and retrieving massive multi-dimensional arrays, respectively. As stated by Baumann et al. [65] (p. 92), the scalable RasDaMan "enables direct interaction, including 3D visualization, what-if scenarios relying on the open OGC 'Big Geo Data' standards suite, the Web Coverage Service (WCS)". Additionally, the Geography Markup Language (GML), Web Map Service (WMS), and Web Coverage Processing Service (WCPS) standards are supported. A RasDaman driver is available for GDAL and MapServer, and RasDaMan also provides tertiary storage archive systems to manage and handle thousands of tertiary storage media, such as magnetic tapes [66]. RasDaMan also provides spatial indexing, adaptive tiling for fast data access, as well as geospatial functions to perform zonal, focal, local, and global raster operations for both regular and irregular raster datasets. The Python RasdaPy client API for RasDaMan requires NumPy, *grpcio* (Python gRPC interface), and *protobuf* (Google's Protocol Buffers) to build and execute RASQL queries. Hein and Blankenbach [67] compare the DBMS RasDaMan with PostgreSQL. Whereas PostGIS incorporates an extension for spatial big data applications, RasDaMan stands out in terms of performance, flexibility, scalability, and fast access to data.

Figure 5 illustrates the schematic representation of the architectural design for the C-SEUSA separated into the presentation, application, and the data layers. SEUSA as a Service represents a Web-GIS application and provides access to toolsets to perform the parallel and distributed geoprocessing operations, upload the use cases, and retrieve and visualize the computational results. The load balancer distributes the workload to the collection of pods according to the respective workload. Pods represent a collection of

containers and are the smallest deployable units within in Kubernetes cluster. A ReplicaSet is responsible for meeting the specifications to maintain the number of replicas of pods. The open-source frameworks such as gRPC, DASK, and RasDaMan are executed within containers to exchange information and spatial datasets. The RasDaMan tiling service partitions the whole array, storing the raster datasets' values into aligned or arbitrary tiles, depending on global or local weighting approaches. Those tiles are in a binary format and structured according to the predefined protocol buffer. Python–Dask obtains the deserialized NumPy arrays and meta-information (e.g., type of decision rule), which are necessary to perform the parallel and distributed SEUSA computations. The SEUSA surfaces, separated into average, min, max, standard deviation maps, and first-order ($S_i$) and total-order ($ST_i$) sensitivity index maps, are returned according to the interface defined in the *.proto* file and stored in the array database. The spatial datasets are accessible through a mapping engine such as MapServer to provide interactive mapping functionalities and are incorporated in the SEUSA as a Service application.

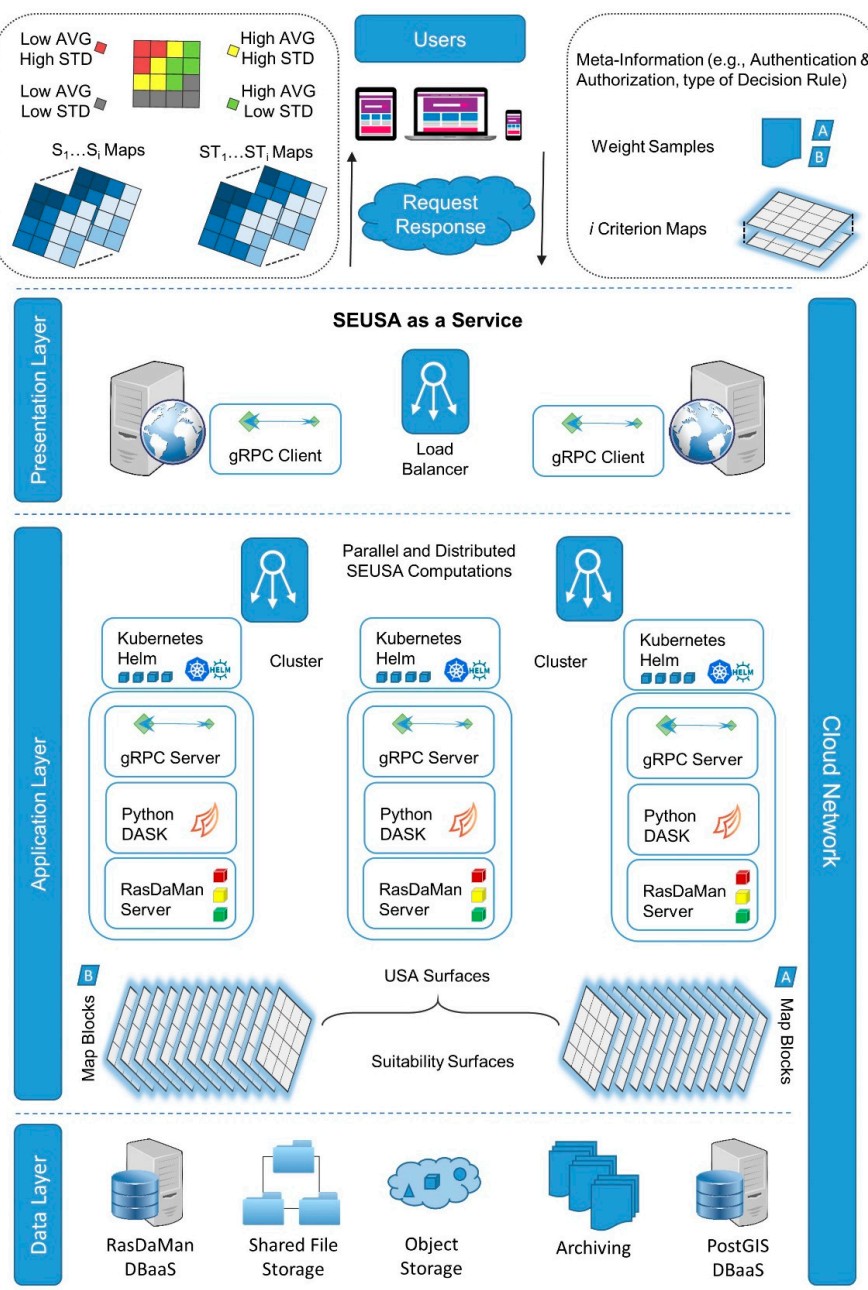

**Figure 5.** SEUSA as Service. Design of the Cloud-based SEUSA framework.

## 4. Discussion

The integration of uncertainty and sensitivity analysis provides a robust approach and methods for identifying and reducing uncertainties, verifying the stability and robustness of the model results, and simplifying model complexity. In turn, reducing model uncertainty and identifying its sensitive parameters and variables improves the quality of the decision-making process in various spatial application domains, such as risk assessment [68,69], ecological vulnerability assessment [70], soil and water assessment [71], land-use change models [72], and disease transmission models [73]. Researchers active in the field of uncertainty and sensitivity analysis implemented and made available software components, packages, frameworks, and tool sets incorporating different uncertainty and sensitivity methods for user communities. For example, *SIMLAB 4.0* is a free software framework for uncertainty and sensitivity analysis. It contains various techniques to conduct sequential global sensitivity analysis, such as Monte Carlo and sampling-based methods [74], developed by the Joint Research Centre of the European Commission. Those methods are accessible through the *R* environment, and the graphical user interface implemented in C# for the .NET framework running on a 64-bit Windows platform. *SALib* is an open-source Python library for performing global sensitivity analysis methods comprising, among others, Sobol Sensitivity Analysis, Fourier Amplitude Sensitivity Analysis (FAST), and Method of Morris [75]. The *R* package *sensitivity* [76] provides a collection of functions, including factor screening, global sensitivity analysis, and robustness analysis. The majority of functions implemented in the package can only be applied to models referring to scalar output. Kc et al. [77] presented a cloud-based framework to perform a computationally intensive and time-consuming sensitivity analysis for wildfire models, comprising input parameters such as local weather conditions, land coverage type, local topography, and fuel conditions. This framework represents a Spark implementation and uses the OpenCL framework for parallel computing and the *SALib* Python library supporting the Method of Morris, variance-based SA methods, and the FAST method.

A few studies have elaborated on Spatially-Explicit Uncertainty and Sensitivity Analysis (SEUSA) in spatial multi-criteria models [8,11,15–19,78,79]. Performing SEUSA for such models incorporating massive and high-resolution spatial datasets for large areas is computationally demanding and memory-intensive, which exceeds the capabilities of a typical workstation, a local cluster, and traditional server applications. This often leads to a compromise between the number of simulation runs and obtaining reliable results for the first-order and total-order sensitivity estimates. SEUSA as a Service for spatial multi-criteria models in a cloud-based environment overcomes those limitations, providing scalable storage, network, and computing resources.

This paper has presented the development of a design for a cloud-based framework to perform Cloud-based Spatially-Explicit Uncertainty and Sensitivity Analysis in spatial multi-criteria models. The framework refers to container virtualization based on Kubernetes, runnable on a variety of cloud platforms such as AWS, OpenStack, GCP, OpenShift, and Microsoft Azure. The framework is not limited to a specific cloud service model, and additional multi-criteria evaluation methods can be implemented for the parallel and distributed SEUSA as a Service. For example, adding further decision rules, such as Ordered Weighted Averaging (OWA), requires extending the gRPC–Dask interface definition and implementing the decision rule. This decision rule incorporates a second set of weights for defining the level of risk and trade-off between the evaluation criteria. Therefore, a message concerning the second weight input dataset and another message for the request must be added to the protocol buffer. The modules to serialize and deserialize the extended protocol buffer content will be available for the client and server stub. The integration of local multi-criteria evaluation approaches incorporating the computation of criterion weights for local neighborhoods requires arbitrary tiling, which is supported by the RasDaMan datacube engine. Supporting this kind of multi-criteria evaluation approach needs an additional enhancement of the current gRPC–Dask interface definition and the implementation of those methods. Testing various S-MCDA uses cases (various decision

rules, different shapes of the arrays reflecting the number of criteria, and number of pixel locations) is necessary to achieve a balanced distribution of the workload. The proposed framework for cloud applications addresses raster-based spatial use cases and does not support vector-based spatial multi-criteria problems at present [11,15]. Rows indicate the alternatives of the vector datasets (e.g., points, polylines, and polygons) and columns refer to the evaluation criteria. Alternatively, representing vector datasets can therefore also be described in the form of multi-dimensional arrays. Consequently, the support of vector-based spatial multi-criteria problems requires an extension of the implemented gRPC interface definition of the proposed SEUSA as a Service application. Alternatively, vector-based scalable cloud approaches, such as GeoRocket [20], might offer an additional opportunity to store, access, analyze, and share vast amounts of geospatial data. It is schema-agnostic and therefore supports a great variety of various geospatial data formats and provides splitting and indexing functionalities to increase the usability, performance, and scalability. The consideration of spatio-temporal uncertainty and sensitivity analysis approaches [80] and different sampling techniques (e.g., Latin hypercube sampling (LHS), replicated LHS, and winding stairs) and calculation methods (e.g., random balance designs, FAST, and extended FAST) [12] also represents another important aspect of extending the capabilities of the SEUSA as a Service for a broader spectrum of spatial use cases. According to Iosifescu-Enescu, Matthys, Gkonos, Iosifescu-Enescu, and Hurni [62] (p. 11), secure serverless cloud architectures for Web-GIS applications, such as SEUSA as a Service, should comprise a managed firewall, security assessment, and a managed Distributed Denial of Service (DDoS) protection, which represents another important aspect for the deployment. SEUSA as a Service is intended to be available to scientists and research communities, and be accessible through a web application in a cloud-based environment capable to perform uncertainty and sensitivity analysis for user-specific use cases in the context of spatial multi-criteria analysis. It will provide data preparation and visualization functionalities and scalable geoprocessing and tiling services, depending on the requested workload. Furthermore, we plan to publish the source code of this framework through online repositories (e.g., GitHub) to offer different research groups the opportunity to deploy their services in private cloud environments. This approach follows the idea of reproducibility and lays the foundation for further developments of the framework.

## 5. Conclusions

As stated by Krämer [20] (p. 5), there is an ongoing paradigm shift in informatics and geoinformatics in particular. Cloud computing leads to significant advancements in application design and development, resulting in high availability, reliability, and scalability of spatial and non-spatial applications. Therefore, the migration of the parallel and distributed SEUSA approach to the cloud represents an attractive opportunity to increase the applicability for highly computationally demanding and complex spatial use cases. This paper illustrates the design of a framework to conduct SEUSA as a Service in a cloud-based environment, representing a scalable, extendable, and applicable solution for massive raster datasets. We posit that SEUSA as a Service will enable exploration and a better understanding of spatial multi-criteria decision analysis results and therefore contribute to improved quality of decision-making. The research presented in the paper is a step in developing quantitative measures for assessing the robustness of complex S-MCDM model solutions and increasing the availability of SEUSA for a broader research community.

**Author Contributions:** Conceptualization, Christoph Erlacher and Karl-Heinrich Anders; methodology, Christoph Erlacher, Karl-Heinrich Anders (parallel and distributed geo-computations) and Piotr Jankowski (SEUSA workflow); writing—original draft, Christoph Erlacher; visualization, Christoph Erlacher (figures and table); writing—review and editing, Piotr Jankowski, Karl-Heinrich Anders, Thomas Blaschke, Gernot Paulus and Christoph Erlacher; supervision and project administration, Gernot Paulus. All authors have read and agreed to the published version of the manuscript.

**Funding:** Piotr Jankowski's work on the paper was supported by the National Science Centre (Narodowe Centrum Nauki, Agreement No. UMO-2018/29/B/ST10/00114).

**Conflicts of Interest:** The authors declare no conflict of interest. The funders had no role in the design of the study; in the collection, analyses, or interpretation of data; in the writing of the manuscript, or in the decision to publish the results.

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
