# Peer review of "A Framework for Cloud-Based Spatially-Explicit Uncertainty and Sensitivity Analysis in Spatial Multi-Criteria Models"

_ijgi, doi:10.3390/ijgi10040244_

Round 1

Reviewer 1 Report

The paper is about designing a framework for sensitivity analysis of the raster data set, with a particular focus on scalability in terms of the amount of data that it can handle.

The paper is well-written, technically sound, and well-structured. The only clarification, maybe, is: how this service is available to other researchers. It will be nice if the authors provide a plan of the deployment and access.

Author Response

Dear Reviewer,

Thank you for your positive feedback. You can find the response to your comment in the attached file.

Kind Regards,

Christoph Erlacher

Reviewer 2 Report

None

Author Response

Dear Reviewer,

Thank you for evaluating our manuscript.

Kind Regards,

Christoph Erlacher

Reviewer 3 Report

The present manuscript is dedicated to cloud-based spatially explicit uncertainty and sensitivity analysis. A new framework is presented that was developed by researchers from different renowned universities. In my opinion, this paper should definitely be published, as it presents general thoughts on establishing cloud-based approaches to the fields of uncertainty and sensitivity analysis. Before final acceptance, authors should strengthen their discussion chapter. The discussion rather reads like a summary, and there are only a few connections made with current state-of-the-art literature. The author teams should expand the discussion and make clear the innovation, compared to other current projects.

Author Response

(The authors gave the same response as above.)

Round 2

Reviewer 3 Report

The author team prepared a revised manuscript version. This new version considers the aspects mentioned in review round no. 1. I was a bit surprised that the response letter only points to the changes but does not discuss the ideas behind the changes in detail. In total, I would recommend this manuscript for publication in this impact journal.